# Transposable Elements Co-Option in Genome Evolution and Gene Regulation

**DOI:** 10.3390/ijms24032610

**Published:** 2023-01-30

**Authors:** Erica Gasparotto, Filippo Vittorio Burattin, Valeria Di Gioia, Michele Panepuccia, Valeria Ranzani, Federica Marasca, Beatrice Bodega

**Affiliations:** 1Fondazione INGM, Istituto Nazionale di Genetica Molecolare “Enrica e Romeo Invernizzi”, 20122 Milan, Italy; 2SEMM, European School of Molecular Medicine, 20139 Milan, Italy; 3Department of Biosciences, University of Milan, 20133 Milan, Italy; 4Department of Clinical Sciences and Community Health, University of Milan, 20122 Milan, Italy

**Keywords:** transposable elements, DNA evolution, retrotransposition, gene regulation, TE de-regulation in diseases

## Abstract

The genome is no longer deemed as a fixed and inert item but rather as a moldable matter that is continuously evolving and adapting. Within this frame, Transposable Elements (TEs), ubiquitous, mobile, repetitive elements, are considered an alive portion of the genomes to date, whose functions, although long considered “dark”, are now coming to light. Here we will review that, besides the detrimental effects that TE mobilization can induce, TEs have shaped genomes in their current form, promoting genome sizing, genomic rearrangements and shuffling of DNA sequences. Although TEs are mostly represented in the genomes by evolutionarily old, short, degenerated, and sedentary fossils, they have been thoroughly co-opted by the hosts as a prolific and original source of regulatory instruments for the control of gene transcription and genome organization in the nuclear space. For these reasons, the deregulation of TE expression and/or activity is implicated in the onset and progression of several diseases. It is likely that we have just revealed the outermost layers of TE functions. Further studies on this portion of the genome are required to unlock novel regulatory functions that could also be exploited for diagnostic and therapeutic approaches.

## 1. Introduction

In the 1940s, the geneticist Barbara McClintock discovered the transposons in maize, i.e., DNA elements that can change location in the host genome. For these elements, she further suggested the role of “controlling elements” as factors that can influence gene expression accordingly to their insertion position in the genome [1,2]. McClintock’s studies overturned the concept of the genome as a fixed and inert item in somatic cells, turning it into alive matter [3,4]. Although ignored for a long time, her discoveries set the basis for the study of how Transposable Element (TE) mobilization has contributed to the evolution of DNA sequences and the regulation of genome functions [5,6]. In respect of these topics, many works described the effects and influence of TEs on genome biology and its regulation; in the current review, we aim to review these findings in order to shed light on how TEs have co-participated in the evolution and adaptation of organisms by increasing their genome plasticity. Briefly, we will show how TEs have colonized genomes of a multitude of species, from bacteria to humans, via retrotransposition, vertical inheritance and horizontal transfer [7,8]. Through their mobilization, TEs have shaped the genomes of complex organisms in their current form, and their presence has been co-opted by the host genomes evolving novel regulatory functions. Finally, we will provide evidence that the misregulation of TE activity accounts for pathological states and disease progression.

## 2. The Biology of Transposable Elements

### 2.1. TE Classification and Distribution in the Human Genome

TEs are interspersed DNA repetitive elements that occupy roughly 47% of the human genome [9,10,11]. Based on their mechanism of transposition, TEs can be classified as DNA transposons (3% of the human genome) that move by a “cut and paste” mechanism [12] or as retrotransposons that adopt a copy and paste mechanism exploiting their RNA as an intermediate [13,14]. Reaching almost 44% of the human genome coverage, retrotransposons comprise the majority of TEs [9] and are additionally classified in LTR (Long Terminal Repeats) or non-LTR [7]; these subclasses are subsequently categorized into different superfamilies, families and subfamilies based on the phylogenetic origins and sequence homology [15,16,17,18].

In humans, LTR retrotransposons are represented by human endogenous retroviruses (HERVs), distributed in class I, II and III, and MaLRs. HERVs cover ~9% of the human genome [19,20,21], are ~9.5 kb long elements, and are derived from exogenous retroviruses [20,21].

Non-LTR retrotransposons include Long Interspersed Nuclear Elements (LINEs, ~22% of the human genome), Short Interspersed Nuclear Elements (SINEs, ~13% of the human genome) and SINE-R/VNTR/Alu (SVA, ~0.15% of the human genome) [9,15,16,22]. LINEs comprise LINE1, LINE2 and LINE3. LINE1 (often called L1) are ~6 kb long and are the only elements still active and autonomous in humans, reaching 18% of genome coverage [10,23,24]. In humans, SINEs are prevalently Alu, sequences of ~300 bp [9,25]. LINE1 and Alu elements range from 500,000 to more than 1 million copies, being the most abundant superfamilies that extensively constellate the human genome [10,26]. SVA represents the most recent family of active, non-autonomous retrotransposons, whose elements are hominid-specific, ~2 kb long and present in less than 3000 copies [27,28,29].

### 2.2. Retrotransposons Can Mobilize in the Genome

In the human genome, the most abundant superfamily of non-LTR retrotransposons is LINE1 [10]. LINE1 is the only autonomous and still active TE, owning the necessary genetic information for its retrotransposition machinery [30,31]. A full-length LINE1 element is approximately 6 kb long and is composed of a promoter responsive to RNA Pol II located in the 5′UTR, two open reading frames encoding for ORF1 (with a chaperon activity) and ORF2 (which has reverse-transcriptase and endonuclease activity) [32,33,34], and 3′UTR followed by a poly-A tract. LINE1 integrates into the genome through a mechanism called target-primed reverse transcription [14,35] (Figure 1). Once a LINE1 element is transcribed, the mRNA is exported to the cytoplasm (Figure 1A), where it is translated into ORF1 and ORF2 proteins [36]. These proteins preferentially bind to the corresponding coding RNA forming a ribonucleoprotein particle (Figure 1B) [36]. The ribonucleoprotein particles then shuttle to the nucleus [37], where ORF2 protein nicks preferentially a 5′-TTTT/AA-3′ DNA sequence (Figure 1C). The TTTT/AA DNA sequence is instrumental for the complementary pairing with the LINE1 RNA poly-A tail; in this way, a 3′OH group of the LINE1 RNA tail is exposed and used to prime the reverse transcription of the element (Figure 1D) [33,38]. Further, LINE1 ORF2 protein also performs the cleavage of the second strand and completes the synthesis of the element [39]. However, this process is inefficient and often results in truncations at the 5′ end, creating a new copy that, lacking the promoter and coding potential, cannot further retrotranspose (Figure 1E) [22]. Interestingly, the LINE1 retrotransposition machinery can be exploited for the mobilization of non-autonomous elements, such as Alu and SVAs, and less frequently by other cellular RNAs as some small nucleolar RNAs (snoRNAs) and small nuclear RNAs (snRNAs), and occasionally by pseudogenes [35].

HERVs were active approximately 5 million years ago; they derive from retroviruses probably engulfed in the progenitors of the host species and integrated into descendants’ genomes by vertical inheritance. To date, in humans, they are considered immobile elements due to mutations and degeneration of their sequence [40]. However, here we will briefly describe the passages of the HERVs mobilization cycle. Genetically, active HERV elements should be endowed with *gag*, *pol* and *env* genes [40,41] and LTR [42]. The pol protein is reverse transcriptase and integrase, while gag generates capsid particles [40,41]. HERVs reverse transcription should have occurred in the cytoplasm, employing host tRNA molecules as primers [43]; the cDNA assembled with the viral-like particle was the intermediate that mediates the integration of the cDNA molecule in the genome [43]. Differently from retroviruses, HERV elements do not code for functional env protein, so they are not able to produce infective particles.

It is well documented that a plethora of organisms experienced waves of TE mobilization that contributed to their genomic re-sizing and shaping [5,15]. Retrotransposon expression and mobilization are tightly regulated events that can be part of specific developmental programs, as they occur in embryogenesis and neuronal differentiation to increase genome plasticity and cellular diversifications [24,44,45,46,47,48]. On the contrary, TE de-regulation is now considered an important player associated with disease onset and progression, as it is demonstrated for genetic diseases, cancer, inflammation, and neurological disorders [49,50,51,52]. Furthermore, TEs can further impact the host genome through transduction mechanisms: during their mobilization, TEs may transfer adjacent genetic material distinct from their intrinsic sequence. In detail, throughout LINE1 and SVAs transcription, the stop signals at the 3′ end may be skipped, provoking the lengthening of the transcription to downstream sequences. These sequences can be occasionally inserted together with the TE in novel genomic positions. A compelling example is the acyl-malonyl condensing enzyme 1 (*AMAC1*) gene family generated by multiple SVA-mediated 3′ transductions on different chromosomes [53].

## 3. How TEs Promote Genome Evolution beyond Retrotransposition

### 3.1. Horizontal Transposon Transfer Is an Ancestral Mechanism of DNA Motion

TEs are able to colonize host genomes by exploiting Horizontal Transposon Transfer (HTT) [8,54]. Horizontal transfer (HT) refers to the transfer of genetic material in the absence of asexual or sexual reproduction [54]. This genetic transfer can even cross inter-species boundaries [55].

The events of HTT are challenging to recognize and study for two main reasons: TE homologies among species, which make arduous the identification of TE provenience, and TE degeneration, which renders their evolutionary tracking more cryptic and blurred [55]. Therefore, horizontal transfer events that involve TEs are mostly assessed according to three long-standing parameters: (i) inconsistency between host and TE phylogenies, (ii) elevated TE resemblance in distantly related species, (iii) uneven distribution of TEs among phylogenies [55,56]. Recently, novel comparative measurement criteria have been introduced to increase the accuracy of HTT evaluation, even though the integration of multiple methodologies remains the best approach [57,58].

The recent advent of omics technologies has allowed us to trace back HTT events among taxa and annotate them in a public HTT-DB database [59,60]. In addition, to browse genomes and infer HTT events, Wallau and colleagues devised the first statistically supported software that discerns horizontal from vertical TE-transfer and predicts putative donor and acceptor species; it is available as an R package named VHICA (Vertical and Horizontal Inheritance Consistence Analysis). VHICA identified 24 HTT of DNA transposon in 20 Drosophila genomes through synonymous substitutions and codon usage bias analyses [61].

Although it was clear that HTT follows different routes from those of the parent-to-offspring inheritance, in which genetic material is vertically inherited, the precise mechanisms by which inter-species TE propagation occurs were initially elusive. To date, there is growing evidence that viruses and other vectors could be the principal catalysts [54]. More in detail, HTT may be initialized by viruses already integrated into the host genome, as it is described for poxviruses, whose re-activation may lead to viral and TE sequences excision, encapsulation and spreading among susceptible unrelated species [62]. Upon infection and viral cycle initiation, both the TE and the pathogen’s genetic material are re-integrated into the DNA of the novel host (Figure 2A) [54]. Besides viruses, it is well known that endosymbiotic bacteria can serve as HTT vectors since they can carry TEs, transferring and integrating them with their genome into a different one [63]. Furthermore, pathogens that can infect gonads may foster the parent-to-offspring transmission of the foreign TE via the HTT mechanism [64].

The first example of eukaryotic HTT dates back to the 1980s with the identification of *Drosophila willistoni* species as a donor of DNA transposon P elements by horizontal transfer into *D. melanogaster* [65]; after that, Baculoviruses were described as the principal HTT mediators among moth species [64,66]. Lately, different groups proposed that ancient HT events served as grafting of the most abundant TE families present in eukaryotic genomes: LINE1 and Bovine-B (Bov-B, long interspersed nuclear elements), a phenomenon mediated by specific parasites [67,68]. Although HTT occurrence is higher in closely related species, there are studies documenting the TE conveyance between the animal and plant kingdoms [69,70]. In 2016, Lin and colleagues provided strong evidence supporting the first HTT spanning flora and fauna, finding that a type of Penelope-Like Retroelement was anciently transferred from arthropods to conifers populating different downstream lineages [69].

Developing novel methods and integrative computational algorithms will help update and refine the phylogenetic trees by adding horizontal connections beside verticals. The full understanding of HTT phenomena could elucidate at a deeper level how TEs have contributed to genomes expansion in evolution.

### 3.2. Mobile Elements Induce DNA Structural Rearrangements

Since TEs are highly homologous, they can impact genome form also by inducing structural variations, a phenomenon that is independent of their transposition. Sequence similarity and 3D space proximity between two transposons can give rise to TE-mediated recombination events that consist of crossing over between non-allelic homologous elements (Figure 2B) [71]. These recombination events involve LINE1 and Alu and can determine deletions, duplications, inversions, or translocations of DNA [6,72]. For instance, LINE1 and *Alu* contributed to nearly half of the human and chimpanzee genomic inversions and demarcated species-specific genotypic and phenotypic variations [73]. More recently, Pascarella et al. highlighted that non-allelic homologous recombination (NAHR) between the youngest LINE1 and Alu elements generates genome diversity in human somatic cells in the developmental process or pathological context [71]; these events occur upon double-strand break repair. Importantly, this study has described that NAHR between homologous LINE1 or Alu elements occurs in a tissue-specific fashion and contributes to genome diversity in development, as stated for neuronal differentiation. Alterations of these genomic events have been described to be associated with Parkinson’s and Alzheimer’s diseases [71,74].

Occasionally, TE insertions can concomitantly result in genomic deletions [6], which can be generated by the independent cutting activity of TE endonucleases or by LINE1 ORF2 protein reported to induce genomic gap formation at particular neuronal stages [75].

In conclusion, mobile elements promote genomic rearrangements, genome sizing and TE-unrelated sequence shuffling, a remodeling that can contribute to genomic innovation and organism adaptation [5,6].

## 4. Transposable Elements Orchestrate Cell Identity and Genome Plasticity

### 4.1. TEs Articulate the Transcriptional Landscape of Host Genomes

Since Barbara McClintock’s discoveries [1,2], the grasp of TE diversity and function has widened and their functional roles as a source of genetic variability and novel regulatory functions keep coming to light [76].

It is well known that the content and variety of TEs are species-specific [7]; however, several TEs have been conserved for millions of years, even among unrelated species, such as in vertebrates [77]. Indeed, although most TEs exist in modern genomes as short fossils, they still contribute as regulatory sequences to the cell transcriptional regulatory network [78,79,80,81]. The co-option of TEs for multiple regulatory functions [79,82,83] is also called exaptation [84]. To name some examples, TEs can provide regulatory sequences as promoters [85], enhancers [80], transcription factor binding sites (TFBSs) [86,87], splicing sites [88], polyadenylation signals [89] and can generate novel regulatory transcripts for gene expression control [90] (Figure 3A–D). More in detail, HERVs contain RNA Pol II promoters in their 3′ and 5′ LTRs [91], while LINE1 includes an internal RNA Pol II promoter [92] and an antisense promoter [93] in 5′ UTR regions. As a result of the endogenous presence of cis-regulatory sequences, the insertion of a transposon, if occurring in the proximity of a coding gene, can impact its expression in different ways [79,85,94].

In 2009, a FANTOM consortium’s study showed that, in humans and mice, 6–30% of RNA transcripts originate from a repetitive element and that retrotransposons, located in proximity to genes, are often used as alternative promoters, discovering roughly 250,000 retrotransposon-derived TSS [95]. Within this frame, we can mention the human amylase (*AMY1C*) gene, whose expression is restricted to parotid tissue because of an endogenous retroviral sequence (*ERVA1C*) acting as a tissue-specific promoter [96]. Similarly, endometrial expression of decidual prolactin (*dPRL*) is regulated by two DNA transposons, MER20 and MER39 [97]. Furthermore, in cancer development, the re-activation of cryptic TE-promoters to regulate the widespread expression of oncogenes has been thoroughly described, a phenomenon also called “onco-exaptation” [98,99]

Regarding TE exaptation as an enhancer, it has been described that TEs exert crucial functions in mammalian development, as in pre-implantation and at the fetal maternal interface [100], morphogenesis and cellular differentiation. In detail, ERV-derived enhancers have been described to regulate genes involved in placenta development [100] and mammary gland morphogenesis [101]. Furthermore, TE-derived enhancers act in the development of mammalian-specific structures such as the neocortex [102] and forebrain neuron patterning [103]. In addition, TE-derived enhancers have been described in the context of inflammation signaling pathways, where ERVs are present in numbers of *IFN*-inducible enhancers [21,104] (Figure 3A) and in the regulation of circadian genes, where murine-specific SINE2, called RSINE1, are binding sites for circadian regulators (as BMAL1, CLOCK) [105].

TEs can also contribute to cell regulatory networks by regulating gene expression at the post-transcriptional level, as demonstrated for Alu sequences involved in the staufen1 (STAU1)-mediated mRNA decay (SMD). STAU1 can induce RNA degradation by recognizing double-stranded RNAs generated from the pairing of Alu sequences retained both in the 3′UTR of an mRNA and in a cytoplasmic long non-coding RNA (Figure 3B) [106].

TEs can also supply alternative splicing sites; it has recently been described that tail-loss evolution in hominoids is due to an AluY insertion into the *TBXB* gene that has resulted in an alternatively spliced isoform sufficient to the tail-loss phenotype of modern hominoids [107]. Similarly, LINEs were discovered to regulate alternative splicing and the exonization of tissue-specific exons. Evolutionarily young LINEs are repressive by recruiting MATR3, PTBP1, and HNRNPM, preventing cryptic splicing of intronic regions; by contrast, older LINEs are more likely to lose the binding sites of repressive splicing proteins and are spliced as exons in a tissue-specific manner (Figure 3C) [88].

TE transcription, when occurring concomitantly to that of neighboring genes, can generate novel non-coding RNAs [95] and chimeric RNAs, both in sense and antisense [78,79,98,108]; but it can also be part of a specific developmental program where the TE-containing transcripts play important regulatory functions. In Percharde et al., RNA transcribed from LINE1 was discovered to actively regulate gene expression in embryonic stem cells (ESCs) by promoting the exit from the 2-cell stage alongside ESC self-renewal [109]. In human adult cells, we have recently identified that quiescent T lymphocytes accumulate at chromatin novel transcript variants generated by the splicing of intronic, evolutionary old LINE1 to keep the expression of genes required for T cell activation and effector function paused [110].

Altogether, these results robustly support the role of TEs as a prolific source of innovation for the regulation of gene expression.

### 4.2. TEs Guide the Three-Dimensional Genome Organization

TE domestication has evolved likewise to contribute to the genome’s higher-order folding. Insulators are regulatory DNA elements that, through CTCF and cohesin binding, organize eukaryotic chromatin by blocking enhancer-promoter interactions and exerting a barrier activity that promotes topological associating domains (TADs) establishment. There are several examples of insulator sequences containing TEs, as binding sites for CTCF and cohesin originating from species-specific retroelement expansion [111,112,113]. In several mammals, comprising rodents, dogs and opossums, CTCF-bound DNA motifs include tRNA-derived SINEs elements [112,114].

Apart from being a source of structural motifs for nuclear architecture, TEs can temporarily orchestrate genome accessibility and organization via their transcribed RNAs. One of the first studies highlighting the existence of a relationship between repetitive-rich nuclear RNAs and chromatin segregation was provided by Hall et al. in 2014. By using C0T-1 probes enriched for TE sequences, the authors described that highly abundant TE-transcripts localize in the euchromatin of interphase nuclei, revealing a structural role for these RNAs in preserving chromatin accessibility [115]. In the wake of this work, temporal regulation of LINE1 expression emerged as a critical step in tuning chromatin accessibility during developmental progression in early mouse embryos [116]. Furthermore, it has been reported that transcripts derived from TEs play a role in three-dimensional genome organization. The first work in this route suggested that transcribed SINE are sufficient to establish distinct functional genomic domains in mammalian development [112], and more recently, it has been demonstrated that transcribed HERV generate de novo TAD boundaries in human pluripotent stem cells (hPSCs) (Figure 4A) [117].

More intriguingly, it seems that the preferential distribution of TEs across different chromatin compartments is consistent with the host gene’s function, its transcriptional profile, and its 3D organization. SINE repeats are mainly distributed in exons of genes with housekeeping functions; LINE1s strongly decorate genes with specialized functions, whereas LTRs are diffused in genes involved in transcription regulation and developmental processes [118]. SINE and LINE1 repeats sequester their enriched genes in distinct active and inactive nuclear domains, respectively, and LINE-1 RNA further works in sequestering LINE1-enriched genes into inactive NADs (nucleolar-associating domains) and LADs (lamina-associating domains) [118] (Figure 4B).

Finally, several hypotheses claim that heterochromatin evolved as a host defense mechanism to silence TE activity [119,120] and that heterochromatin compartments are formed via phase-separation, where TEs are scaffolds for condensate formation [121,122]. Asimi et al. recently published that, in TRIM28 KO mESCs, the re-activation of ERVs located nearby pluripotency genes sequester RNA Pol II and Mediator coactivator complex into droplets, inducing the disassociation of transcriptional condensates from pluripotency genes and thus their transcriptional repression [123] (Figure 4C).

So far, the versatility and the polyvalence of TEs have turned out to be the key to understanding genome and organism complexity.

## 5. Linking Deregulation of TE Expression and Activity to Diseases

### 5.1. Host Genomes Evolved Sophisticated Strategies to Govern TE Expression and Activity

Retrotransposition can be detrimental to the host organism due to its intrinsic ability to introduce mutations and promote recombination [124,125]. Consequently, a fundamental factor for the success of the mutual coexistence between TEs and the host genome is its ability to minimize novel insertions [126]. Indeed, cells exploit several epigenetic mechanisms to silence retrotransposons, among which DNA methylation (mediated by DNA methyltransferases (*DNMT1* and *DNMT3*)) [127] and deposition of repressive histone marks such as H3K9me2/3 (mediated by G9a, SETDB1 and SUV39H) [128,129]. In particular, the histone methyltransferase SETDB1 generates a complex with TRIM28, which is, in turn, recruited by KRAB-ZNF proteins on TE promoters, inducing TE repression [128,130]. KRAB-ZNF proteins are a notable testimony of the arms race between TEs and host genomes: their evolutionary age correlates with the age of the elements they repress [130,131].

TE retrotransposon activity is also inhibited at the post-transcriptional level through RNA interference machinery by piRNAs that have a crucial role in restricting retrotransposon activity in the germline [132]. Furthermore, PIWI proteins are also able to promote de novo DNA methylation on transposon sequences [133,134].

Although the host genome has evolved different mechanisms to avoid uncontrolled TE expression and mobilization, if tightly regulated, they can be part of specific developmental programs, both in development and in somatic-adult cells, contributing to cellular genome plasticity and functional diversifications [24].

In somatic tissues, as in neuronal progenitor cells (NPC), LINE1 mobilization is partially under the control of Wnt/HDAC1/Sox2 regulation during differentiation [135,136], suggesting a correlation between TE activity and tissue-specific transcriptional program. This phenomenon seems to be instrumental in amplifying the plasticity and cognitive potential of the nervous system by increasing somatic mosaicism in the brain [135,136,137,138,139].

When the tight and precise regulation of TE expression and mobilization is lost, it can lead and contribute to several pathological states as occurs in cancer, neurodevelopmental diseases, aging and inflammatory-based pathologies [49,50,51,52]

### 5.2. Deregulated TE Activity and Expression Drives Pathological States

TEs have been associated with pathological context due to their ability to insert in novel genomic locations and thus disrupt genes [50,140]. More recently, the uncontrolled expression of TEs has been linked to several diseases [49,51].

The first deleterious TE insertion was detected in the Coagulation Factor VIII (*F8*) gene in the germline, where a LINE1 insertion into the exon 14 causes Hemophilia A [141] (Figure 1). Following that, several genetic pathologies were associated with misplaced retrotransposition events in the germline. E.g., Alu insertion in the *POT1* gene was found in Walker–Warburg syndrome patients [142], and LINE1 insertion in the *CYBB* gene was causative of chronic granulomatous disease [143].

In somatic cells, novel LINE1 insertions have been associated with cancer and neurodegenerative diseases [124]. Regarding the former, in colon tissue, the interruption of the *APC* gene by a LINE1 insertion causes colorectal cancer [144,145]. However, de novo LINE1 insertions are frequently found in many different cancer types, with a high recurrence in oncosoppressors, such as *PTEN* [146]. The aberrant mobilization of TEs is also associated with neurodegenerative diseases (i.e., Alzheimer’s and Parkinson’s disease, amyotrophic lateral sclerosis (ALS)) and psychiatric disorders (i.e., schizophrenia, bipolar disorders, autism and attention deficits [49,52,147,148,149,150,151,152,153,154]). In particular, patients affected by Rett syndrome and ataxia telangiectasias display an altered LINE1 copy number in brains that possibly influence the neurodegenerative phenotype and the brain pathophysiology [155,156,157]; while schizophrenic patients have been reported to display altered LINE1 or HERVs expression and or activity in the brain [158,159,160,161]. Moreover, deregulated expression of TEs has been connected to aging and senescence-associated inflammatory phenotype, where the loss of heterochromatin, in turn, promotes TE de-repression [162,163]. In particular, pieces of evidence support that inflammation is promoted by a cytoplasmatic accumulation of nucleic acids, deriving from TE expression/retrotransposition [22,164,165], that promotes IFN-I response [165,166]. This is an ancestral defense mechanism for virus infections [167]. An aberrant accumulation of nucleic acids derived from TEs has also been described to contribute to autoimmune diseases, such as systemic lupus erythematosus and Sjögren’s syndrome. In patients affected by these pathologies, autoantibodies for Ro60 are produced. Ro60 is an RNA binding protein that negatively regulates Alu transcription; in Sjögren’s syndrome, the absence of Ro60 induces Alu re-expressed and causes the upregulation of IFN-I genes [22].

## 6. Conclusions

Barbara McClintock discovered the first transposon in the 1950s in maize, but her findings were ignored for decades until TE sequences were also discovered in bacteria. We waited until 1983 to see her discoveries fully recognized when she was finally awarded the Nobel prize in Physiology or Medicine. Afterward, the release of the first whole human genome sequencing in 2001 highlighted a disproportion in genomic coverage, where protein-coding genes represent only a small fraction of the genome, while, unexpectedly, transposons cover almost half. In the 2000s, the ENCODE and FANTOM international consortia were raised with the main purpose of identifying respectively genomic functional elements and transcriptome complexity. From that moment onward, in the past two decades, important studies have been published, greatly contributing to uncovering the hidden functions of TEs.

Here, we have reviewed the most important findings in the field demonstrating that TEs represent a source of genome innovation with implications for inter- and intra-species diversification. Being able to colonize DNA, TEs have shaped the genomes of the host organisms in their current size and organization. We have also provided an overview of how transposons have been co-opted by the host genome to increase and diversify its functions, modulating gene transcription and genome topology.

The implementation of Next Generation Sequencing (NGS) approaches has been fundamental to understanding the expression and the functions of TEs in gene regulatory networks. However, still, the study of the repetitive genome has several limitations due to their overrepresentation and their sequence homology, characteristics that render TE-reads mappability compelling. Nowadays, the implementation of long reads sequencing [110] and dedicated bioinformatic approaches [168] are progressively leading to increased confidence and awareness in the study of transposons, even at the single-cell level [169]. The improvement of technologies dedicated to TE studies will progressively allow the integration of TE analysis in the diagnosis of multifactorial diseases as well as in the definition of their related molecular mechanisms going in the path of personalized medicine.

In conclusion, TEs contribute to genome plasticity and adaptation: from being considered “junk DNA”, TEs are now rather conceived as a fundamental part of genome complexity, although we have just uncovered the surface.

## Figures and Tables

**Figure 1 ijms-24-02610-f001:**
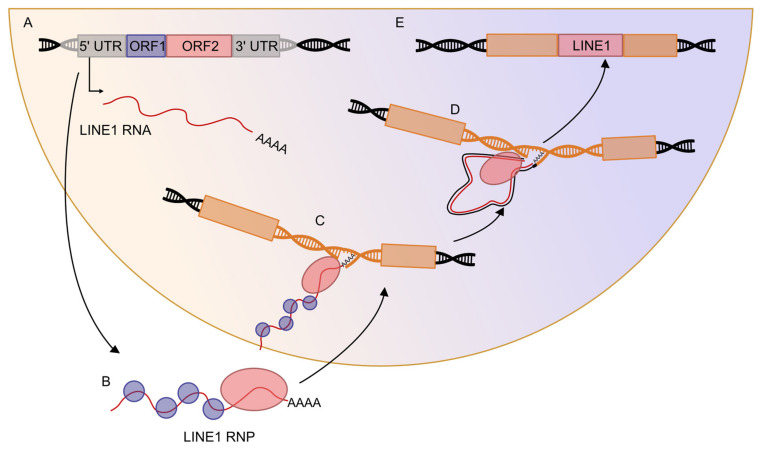
LINE1 retrotransposition mechanism can disrupt gene coding sequence. Mechanism of LINE1 mobilization: (**A**). A full-length retrotransposition competent LINE1 element is transcribed from its promoter and encodes for ORF1 and ORF2 proteins. (**B**). In the cytoplasm, newly translated ORF1 (violet) and ORF2 (pink) bind to their mRNA of origin (red) to form a ribonucleoprotein particle, RNP. (**C**). The RNP complex enters the nucleus after mitosis and targets a novel genomic region bearing the sequence 5′-TTTT/AA-3′, where ORF2 nicks the host DNA. (**D**). ORF2 primes the reverse transcription of LINE1 element. (**E**). The integration of LINE1 can disrupt the coding sequence of a gene, resulting in a dysfunctional one.

**Figure 2 ijms-24-02610-f002:**
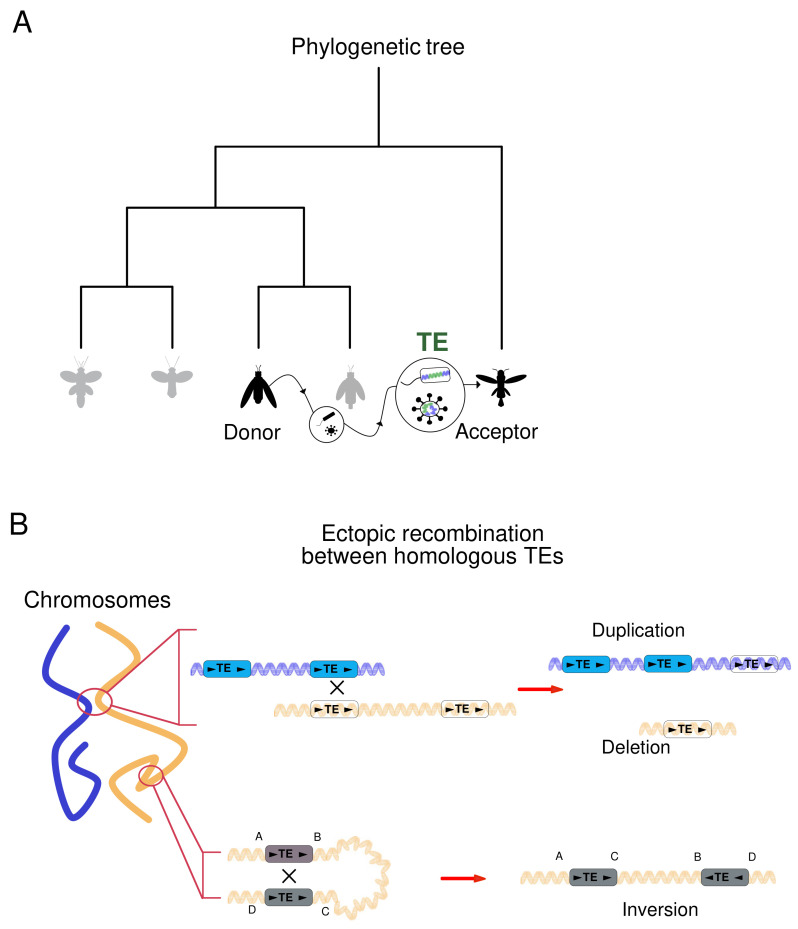
TEs invaded and rewired genomes by Horizontal Transfer and structural variations. Schematic representation of (**A**) a phylogenetic tree highlighting a Horizontal Transposons Transfer between donor and acceptor species. Magnified in circles, putative agents responsible for TEs (green) shuttling, such as bacteria and viruses. (**B**) TEs-mediated ectopic recombination. From left to right: non-homologous chromosomes in close proximity. Ectopic recombination between homologous TEs results in duplication and deletion on the respective chromosomes’ arms. Below, alternative outcome is represented by a genomic inversion mediated by TEs with the structural reconfiguration of the genomic segment affected. Altogether, these dynamic hotspots of DNA recombination contribute to somatic variation, organismal complexity, and diseases [59].

**Figure 3 ijms-24-02610-f003:**
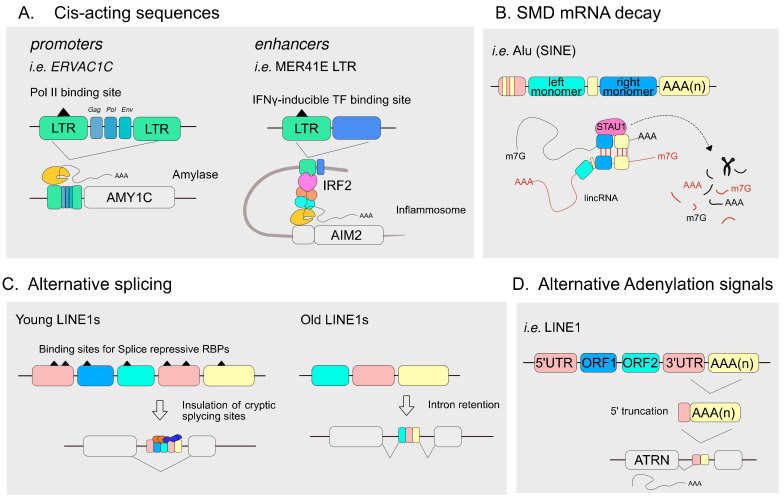
Examples of TEs-mediated regulatory functions. Schematic representation of (**A**) AMY1C regulation owed to the insertion of Endogenous retrovirus ERVA1C that acts as a tissue-specific promoter (left), MER41 carries consensus sequences for transcription factors and acts as an enhancer, recruiting IRF2 to trigger the assembly of AIM2 (right); (**B**) Cytosolic adenylated lncRNAs containing Alu (lincRNA) imprecisely pair to SMD-target poly-adenylated mRNAs to generate dsRNA, recognized by STAU1 protein that initiates the degradation cascade (**C**) Young LINE1 elements harbor consensus sequences for repressive-splice RBPs (right), old evolutionary LINE1s have a higher probability of being intron-retained and generating alternative splicing-isoforms (left). (**D**) The human isoform ATRN derives from a 5′ truncated LINE1 element that retains a stop codon and a polyadenylation signal.

**Figure 4 ijms-24-02610-f004:**
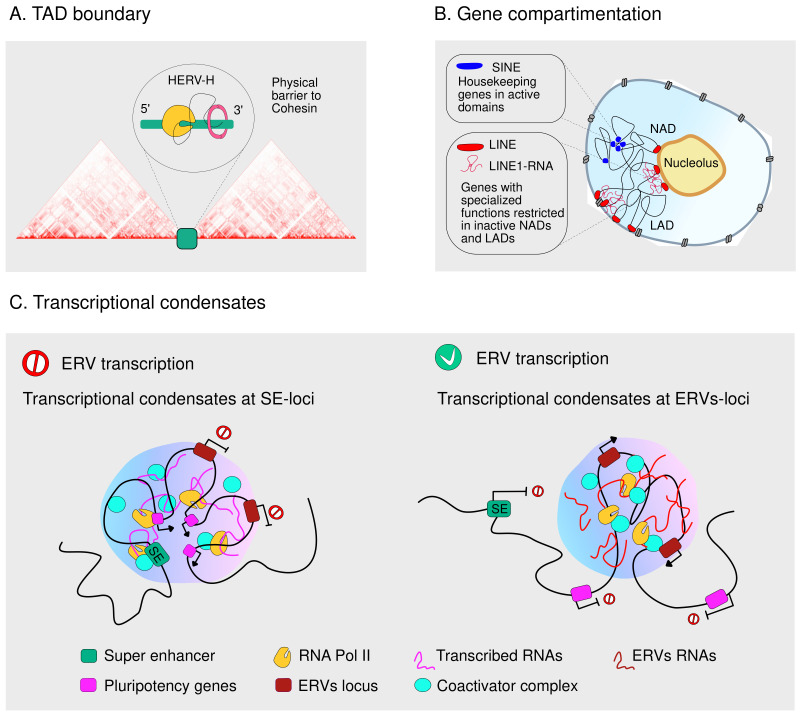
TEs organize the genome in the 3D space. Schematic representation of (**A**) HERV-H transcripts accumulation acting as physical barriers to the cohesin complex’s movement leading to the formation of a TAD boundary; (**B**) TEs guiding gene positioning in respect of their nature; (**C**) TEs RNAs transcriptional condensates: (left) in the absence of ERV active transcription, Pol II and coactivators are recruited at SE loci while, (right) whereby ERV are expressed, Pol II and coactivator complex are redirected into condensates at ERVs-loci affecting the transcriptional output.

## Data Availability

No new data were created or analyzed in this study. Data sharing is not applicable to this article.

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
