# Peer review of "Transposable Elements Co-Option in Genome Evolution and Gene Regulation"

_ijms, 2023, doi:10.3390/ijms24032610_

Round 1

Reviewer 1 Report

The authors present an interesting review on how transposable elements can be co-opted for gene regulation or genome evolution.

I have only minor concerns :

L52 : I am not sure that the reference used to described non-LTR TE is relevant (ref 14).

L54 : I suggest to complete/replace the references for TE classification.

This reference is the first TE universal classification scheme so it should be cited : Wicker, T., Sabot, F., Hua-Van, A. et al. A unified classification system for eukaryotic transposable elements. Nat Rev Genet 8, 973–982 (2007). https://doi.org/10.1038/nrg2165

Then this review allows to see the questions raised following the first TE universal classification scheme : Benoît Piégu, Solenne Bire, Peter Arensburger, Yves Bigot, A survey of transposable element classification systems – A call for a fundamental update to meet the challenge of their diversity and complexity, Molecular Phylogenetics and Evolution, Volume 86, 2015, Pages 90-109, ISSN 1055-7903, https://doi.org/10.1016/j.ympev.2015.03.009.

Figure 1 : the RNP is not defined (ribonucleoprotein particle) and  the letter case for A, B, C D E is not the same beteween the legend and the figure.

L169-171 : I suggest to add  "by providing Transcription Factor Binding Site (TFBS)" as you give an exemple in figure 3C.

L 172 : "Indeed, full-length TEs genetically contain several regulatory regions" The sentence is true but could lead to understand that only full-length TEs can contain regulatory regions which could be co-opted by a nearby gene. While the regulatory sequences of old TEs (TE fragments) can also be co-opted. See as exemple this publication : Baud, Agnès; Wan, Mariène; Nouaud, Danielle; Francillonne, Nicolas; Anxolabéhère, Dominique; Quesneville, Hadi. Traces of transposable elements in genome dark matter co-opted by flowering gene regulation networks. Peer Community Journal, Volume 2 (2022), article no. e14. doi : 10.24072/pcjournal.68. https://peercommunityjournal.org/articles/10.24072/pcjournal.68/

L 273-282 : The conclusion is a bit short and could be enrriched, some perspectives would be welcome.

Author Response

Response to the Reviewer 1 comments on the manuscripts ijms - 2149288

Below we provide detailed answers to Reviewer 1 comments:

Point 1:

  • “ L52 : I am not sure that the reference used to described non-LTR TE is relevant (ref 14)”
  • “L54 : I suggest to complete/replace the references for TE classification. This reference is the first TE universal classification scheme so it should be cited : Wicker, T., Sabot, F., Hua-Van, A. et al. A unified classification system for eukaryotic transposable elements. Nat Rev Genet 8, 973–982 (2007). https://doi.org/10.1038/nrg2165 . Then this review allows to see the questions raised following the first TE universal classification scheme: Benoît Piégu, Solenne Bire, Peter Arensburger, Yves Bigot, A survey of transposable element classification systems – A call for a fundamental update to meet the challenge of their diversity and complexity, Molecular Phylogenetics and Evolution, Volume 86, 2015, Pages 90-109, ISSN 1055-7903, https://doi.org/10.1016/j.ympev.2015.03.009.”

Line 53: (Line 52 in the previous version of the manuscript) we substituted the previous reference with “Wells JN and Feschotte C: A Field Guide to Eukaryotic Transposable Elements. Annu Rev Genet 54: 539-561, 2020”, Ref 7.

Line 55: (Line 54 in the previous version of the manuscript) we added references he/she kindly suggested.

Point 2: “Figure 1 : the RNP is not defined (ribonucleoprotein particle) and  the letter case for A, B, C, D, E is not the same between the legend and the figure”

We have added “ribonucleoprotein particle” to specify the acronym RNP in the legend of Figure 1, and RNP in Figure 1 and amended the labelling.

Point 3:

  • L169-171: I suggest to add "by providing Transcription Factor Binding Site (TFBS)" as you give an exemple in figure 3C.”
  • L 172: "Indeed, full-length TEs genetically contain several regulatory regions" The sentence is true but could lead to understand that only full-length TEs can contain regulatory regions which could be co-opted by a nearby gene. While the regulatory sequences of old TEs (TE fragments) can also be co-opted. See as exemple this publication: Baud, Agnès; Wan, Mariène; Nouaud, Danielle; Francillonne, Nicolas; Anxolabéhère, Dominique; Quesneville, Hadi. Traces of transposable elements in genome dark matter co-opted by flowering gene regulation networks. Peer Community Journal, Volume 2 (2022), article no. e14. doi: 10.24072/pcjournal.68. https://peercommunityjournal.org/articles/10.24072/pcjournal.68/”

We amended as suggested Lines 169– 171 (Line 177-178 in the current version of the manuscript) and Line 172 (Line 179-180 in the current version of the manuscript).

Point 4: “L 273-282: The conclusion is a bit short and could be enriched, some perspectives would be welcome.”

Following the reviewer’s suggestion, we have implemented the conclusion, please refer to Lines 307-314 & 320-327 of the current version of the manuscript.

Reviewer 2 Report

Gasparotto et al. provided a review on the co-option biology of transposable elements (TEs), covering broad research topics such as TE classification, horizontal transfer, genome rearrangement of TEs, gene expression by TEs' regulatory functions, and 3D genome organization organized by TEs. I appreciate that this is a comprehensive and very good review article; however, I raise several points that the authors need to be addressed (see below).

1. I found a number of grammatical errors in the manuscript. The manuscript needs to be proofread by a native English speaker.

2. Many review articles are cited throughout the manuscript. Although I agree that the review papers cited are important, it is recommended that the authors also cite representative original research papers and explain/discuss the findings because the present manuscript itself is a review article.

3. Page 2, line 60; "LINE1 are ~6kb long" --> "LINE1 (L1 )are ~6kb long"

LINE1 may be a formal name, but it is often called "L1".

4. Page 4, lines 120-122; "More in detail, HTT may be initialized by viruses already integrated into the host genome ..."

One of the pieces of evidence has been reported by Piskurek and Okada (doi: 10.1073/pnas.070053110), which should be cited on this page.

5. Page 4, lines 131-132; "Bovine-B retrotransposons" --> "Bovine-B (Bov-B) retrotransposons"

Bovine-B may be a formal name, but it is currently often called "Bov-B". Also, the original work for the Bov-B HTT should be cited (doi: 10.1073/pnas.95.18.10704).

6. Page 4, lines 134; "Liu and colleagues" --> "Lin and colleagues"

7. Page 5, line 164 – Page 6, line 195

As mentioned above, review papers are often cited in this section; however, the authors should consciously cite the representative papers of the original works.

8. Page 5, line 172; "antisense promoters"

The 5' UTR region of LINE1 (L1) has an antisense promoter activity. The authors should mention it in this section. (E.g., doi: 10.1128/MCB.21.6.1973-1985.2001)

9. Page 5, lines 178-180; "TEs elements co-option has been described in the context of inflammation signalling pathways, where ERVs are present in numbers of IFN-inducible enhancers"

This section lacks an important topic on the TE exaptation (co-option) serving as developmental enhancers. Not only in cultured cells, but some TEs are evolutionarily conserved among species and have enhancer activities in developing mammalian embryos (e.g., doi: 10.1038/nature04696, 10.1073/pnas.0709398105). In addition, multiple TEs are known to be bound by a transcription factor, and they could coordinately act as enhancers in a certain cis-regulatory network (e.g., doi: 10.1038/ng.2553, 10.1038/ncomms7644, 10.1093/nar/gkz1003). The authors need to add a discussion about this research background.

10. Page 6, lines 200-202; "There are several examples of insulator sequences containing TEs, as binding sites for CTCF and cohesin originated from species-specific retroelement expansion (76)."

The authors should also cite the important former paper related to this topic (doi: 10.1126/science.1140871).

11. Page 7, lines 221-222; "In this frame, Asimi et al., recently published that TE-derived transcripts can play a function in chromatin compartment organization through phase transition (85) (Figure 4C)."

I do not fully understand the description and the illustration of Fig. 4C. The authors should provide a detailed explanation in the text and/or the legend of Fig. 4C.

Author Response

Response to the Reviewer 2 comments on the manuscripts ijms - 2149288

Below we provide detailed answers to Reviewer 2 comments:

Point 1. I found a number of grammatical errors in the manuscript. The manuscript needs to be proofread by a native English speaker.

The manuscript has been entirely revised for English grammar as requested.

Point 2. Many review articles are cited throughout the manuscript. Although I agree that the review papers cited are important, it is recommended that the authors also cite representative original research papers and explain/discuss the findings because the present manuscript itself is a review article.

The reviewer is right, we have changed accordingly review articles referring to the original article, following her/his suggestion. Please see ref 62; 68; 77; 80-87; 91-94; 96; 98-104; 108; and 112.

Point 3. Page 2, line 60; "LINE1 are ~6kb long" --> "LINE1 (L1) are ~6 kb long"

LINE1 may be a formal name, but it is often called "L1".

We have revised the text following the reviewer’s suggestion, please refer to Line 61.

Point 4. Page 4, lines 120-122; "More in detail, HTT may be initialized by viruses already integrated into the host genome ..." One of the pieces of evidence has been reported by Piskurek and Okada (doi: 10.1073/pnas.070053110), which should be cited on this page.

We thank the reviewer for the suggestion, we have integrated the mentioned citation in the paragraph, please refer to Lines 129-131.

Point 5. Page 4, lines 131-132; "Bovine-B retrotransposons" --> "Bovine-B (Bov-B) retrotransposons"

Bovine-B may be a formal name, but it is currently often called "Bov-B". Also, the original work for the Bov-B HTT should be cited (doi: 10.1073/pnas.95.18.10704).

We have modified the text and cited the suggested paper accordingly to the reviewer's suggestion, please refer to Line 140.

Point 6. Page 4, lines 134; "Liu and colleagues" --> "Lin and colleagues"

We have amended the text accordingly to reviewer’s suggestion, please refer to Line 142.

Point 7. Page 5, line 164 – Page 6, line 195. As mentioned above, review papers are often cited in this section; however, the authors should consciously cite the representative papers of the original works.

The reviewer is right, we have implemented the references cited in the text with original articles, please refer to ref 77; 80-87; 91-94; 96; 98-104 & 108

Point 8. Page 5, line 172; "antisense promoters". The 5' UTR region of LINE1 (L1) has an antisense promoter activity. The authors should mention it in this section. (E.g., doi: 10.1128/MCB.21.6.1973-1985.2001)

We have clarified this point in the text and cited the recommended work (please refer to Lines 179-180 and ref 93).

Point 9. Page 5, lines 178-180; "TEs elements co-option has been described in the context of inflammation signalling pathways, where ERVs are present in numbers of IFN-inducible enhancers"
This section lacks an important topic on the TE exaptation (co-option) serving as developmental enhancers. Not only in cultured cells, but some TEs are evolutionarily conserved among species and have enhancer activities in developing mammalian embryos (e.g., doi: 10.1038/nature04696, 10.1073/pnas.0709398105). In addition, multiple TEs are known to be bound by a transcription factor, and they could coordinately act as enhancers in a certain cis-regulatory network (e.g., doi: 10.1038/ng.2553, 10.1038/ncomms7644, 10.1093/nar/gkz1003). The authors need to add a discussion about this research background.

We thank the reviewer for the suggestion, in the current version of the manuscripts we have briefly described the functional relevance of TE exaptation as an evolutionarily conserved mechanism in developmentally related processes. Please refer to Lines 173-176; 183-185 & 188-202.

Point 10. Page 6, lines 200-202; "There are several examples of insulator sequences containing TEs, as binding sites for CTCF and cohesin originated from species-specific retroelement expansion (76)."
The authors should also cite the important former paper related to this topic (doi: 10.1126/science.1140871).

In the current version of the manuscripts we have added the reference that the reviewer has kindly suggested, please refer to Ref 112, and Line 224.

Point 11. Page 7, lines 221-222; "In this frame, Asimi et al., recently published that TE-derived transcripts can play a function in chromatin compartment organization through phase transition (85) (Figure 4C)."

I do not fully understand the description and the illustration of Fig. 4C. The authors should provide a detailed explanation in the text and/or the legend of Fig. 4C.

We thank the reviewer for her/his comment, we have rephrased the sentence and modified the figure accordingly. Please refer 245-248 & 387-389

Reviewer 3 Report

In this review, Gasparotto et al, present the current literature regarding the role of Transposable Elements (TE) in shaping the genome of their hosts and in regulating the transcription of the host genes.

After a brief introduction to TEs and their mechanism of transposition, they introduce very well the concept of horizontal transfer (HT), as both transposition and HT can be sources of genome evolution. They then describe how TEs can induce DNA structural rearrangements. Subsequently, the authors focus on, what according to the title, is the main subject of the review: how TEs have been coopted by the host genome to fulfill multiple regulatory functions and nicely introduce their identified roles in 3D genome organization. They finish by briefly describing how TEs are deregulated and how their uncontrolled expression and mobilization have been associated with diseases. 

This is an informative review on a topic of great interest and many nice concepts are presented showing the impact of TEs in the genome. The review is well written however, in the title they stress the co-option concept but they do not develop it sufficiently in the main text. In my opinion, the session on TE cooption events and their consequences on human disease should be developed by providing more examples. I have some additional comments that I detail here:

Comments:

Although the English language and style is generally good, there are some imprecisions and a revision is recommended.  For example, in line 15 of the abstract the term “retained” is not used correctly. Furthermore, often an “s“ is added to TE when not appropriate/necessary (e.g. TEs expression which should be TE expression). 

There is a problem with the transition from line 37 to line 38. Are some sentences missing?

In lines 60-61, the authors state that LINEs are the only active elements in humans, but then in lines 63-64, they claim that SVA represents the most recent family of active retrotransposons. I think the authors should distinguish between autonomous and non-autonomous elements and write in line 60 that LINEs are the only active autonomous elements, while SVAs are active but non-autonomous. 

A few more examples of how Transposition can impact the genome need to be added in section 2.2.

In line 148, the authors cite the work of Pascarella et al who showed that TE recombination contributes to genomic diversity. The authors should give more details regarding the results of this paper.  

In line 153, while they seemed to have moved away from transposition, they give an example of how TEs can add genetic material distinct from their intrinsic sequences through transposition. Consider moving to section 2.2.

More examples need to be given and discussed on how TEs have been co-opted by the host genome to fulfill regulatory functions (section 4.1). Additionally, the concept of how TEs can contribute to the evolution of gene regulatory networks should be discussed.

In section 5.2, the authors provide examples of how uncontrolled expression and mobilization of TEs have been associated with diseases. In my opinion, this section is a little short and could be expanded with a few more examples.

Author Response

Response to Reviewer 3 comments on the manuscripts ijms - 2149288

Below we provide detailed answers to Reviewer 3 comments:

Reviewer 3:

The review is well written however, in the title they stress the co-option concept but they do not develop it sufficiently in the main text. In my opinion, the session on TE cooption events and their consequences on human disease should be developed by providing more examples. I have some additional comments that I detail here:

Following the reviewer's advice, we have now significantly implemented paragraph 4.1, lines 183-185 & 188-202 and paragraph 5.2, lines 284-287; 293-296 & 302-304

Point 1: Although the English language and style is generally good, there are some imprecisions and a revision is recommended.  For example, in line 15 of the abstract the term “retained” is not used correctly. Furthermore, often an “s“ is added to TE when not appropriate/necessary (e.g. TEs expression which should be TE expression). 

We have amended imprecisions and the manuscript has been entirely revised for English grammar as requested.
Line 15 (please refer to Line 15 in the current version of the manuscript): we substituted “retained” with “considered”.

Point 2: There is a problem with the transition from line 37 to line 38. Are some sentences missing?

We amended the transition between lines 37 and 38. Please refer to Lines 37-39 in the current version of the manuscript.

Point 3: In lines 60-61, the authors state that LINEs are the only active elements in humans, but then in lines 63-64, they claim that SVA represents the most recent family of active retrotransposons. I think the authors should distinguish between autonomous and non-autonomous elements and write in line 60 that LINEs are the only active autonomous elements, while SVAs are active but non-autonomous.  

The reviewer is right, we have revised the manuscript accordingly to her/his suggestions. Please refer to Lines 62-65 in the current version of the manuscript.

Point 4 i. A few more examples of how Transposition can impact the genome need to be added in section 2.2.

Following the reviewer’s suggestion, we have included a brief overview of the impact of TE life cycle on genome plasticity and functions. Please refer to 95-105 the current version of the manuscript.

ii. In line 148, the authors cite the work of Pascarella et al who showed that TE recombination contributes to genomic diversity. The authors should give more details regarding the results of this paper. 

We thank the reviewer for this advice, in the current version of the manuscript we have now included a brief description of the findings (please refer to Lines 156 – 161).

iii. In line 153, while they seemed to have moved away from transposition, they give an example of how TEs can add genetic material distinct from their intrinsic sequences through transposition. Consider moving to section 2.2.

The reviewer is right, we have moved the mentioned paragraph in section 2.2 (please refer to Lines 100-105).

Point 5: More examples need to be given and discussed on how TEs have been co-opted by the host genome to fulfill regulatory functions (section 4.1). Additionally, the concept of how TEs can contribute to the evolution of gene regulatory networks should be discussed.

We have addressed the issue raised by our reviewer thoroughly implementing section 4.1 with additional examples (please refer to Lines 183-185 & 188-202).

Point 6: In section 5.2, the authors provide examples of how uncontrolled expression and mobilization of TEs have been associated with diseases. In my opinion, this section is a little short and could be expanded with a few more examples.

Following the reviewer’s suggestion, we have added more examples of the involvement of TEs in human diseases, please refer to Lines 284-287; 293-296 & 302-304.
